# Volatile Olfactory Profiles of Umbrian Extra Virgin Olive Oils and Their Discrimination through MOX Chemical Sensors

**DOI:** 10.3390/s22197164

**Published:** 2022-09-21

**Authors:** Roberto Mariotti, Estefanía Núñez-Carmona, Dario Genzardi, Saverio Pandolfi, Veronica Sberveglieri, Soraya Mousavi

**Affiliations:** 1Institute of Biosciences and Bioresources, National Research Council, 06128 Perugia, Italy; 2Institute of Biosciences and Bioresources, National Research Council, URT-Reggio Emilia, Via J. F. Kennedy 17/I, 42124 Reggio Emilia, Italy

**Keywords:** virgin olive oil, aroma, volatile compounds, sensors, local olive cultivars, sensory analysis

## Abstract

Extra virgin olive oil (EVOO) is the best vegetable oil worldwide but, at the same time, is one of the product victims of fraud in the agri-food sector, and the differences about quality within the extra-virgin olive oil category are often missed. Several scientific techniques were applied in order to guarantee the authenticity and quality of this EVOO. In the present study, the volatile compounds (VOCs) by gas chromatography–mass spectrometry with solid-phase micro-extraction detection (GC–MS SPME), organoleptic analysis by the official Slow Food panel and the detection by a Small Sensor System (S3) were applied. Ten EVOOs from Umbria, a central Italian region, were selected from the 2021 Slow Food Italian extra virgin olive oil official guide, which includes hundreds of high-quality olive oils. The results demonstrated the possibility to discriminate the ten EVOOs, even if they belong to the same Italian region, by all three techniques. The result of GC–MS SPME detection was comparable at the discrimination level to the organoleptic test with few exceptions, while the S3 was able to better separate some EVOOs, which were not discriminated perfectly by the other two methods. The correlation analysis performed among and between the three methodologies allowed us to identify 388 strong associations with a *p* value less than 0.05. This study has highlighted how much the mix of VOCs was different even among few and localized EVOOs. The correlation with the sensor detection, which is faster and chipper compared to the other two techniques, elucidated the similarities and discrepancies between the applied methods.

## 1. Introduction

Extra virgin olive oil (EVOO) is a highly valued product in the Mediterranean Diet (MED), and its consumption is increasing around the world, including in countries far from the Mediterranean basin [1,2,3]. In MED, EVOO is the main source of fat since it is composed of a major fatty acid fraction (98–99%), which comprises oleic acid (55–83%) and linoleic acid (2.5–21%) predominantly, and certain minor constituents that include phenolic and volatile compounds, which offer both a multitude of bioactive functions and distinctive organoleptic properties [4,5,6,7,8]. In order to be classified in commercial categories, a sensory analysis of each EVOO is mandatory. This classification is ruled by the European Official Regulations for olive oil (Commission Regulation 640/2008/EC) [9] and is carried out by certified test panels, in which the evaluation of taste (pungency and bitterness) and aroma play a very important role. In this regard, Morales et al. in 1995 [10] developed the statistical sensory wheel (SSW) for virgin olive oil (VOO) to understand the connection between volatile compounds and odor characteristics. The SSW clusters into different categories the sensory attributes of VOO together with the volatile compounds labeled with a particular sensory note, where the green or ripe fruit perceptions are the most important to VOO aroma [11]. EVOO, as “olive juice”, is considered the highest quality vegetable oil, and in general, it is characterized by showing a sensory grade higher than 6.5 points, with a fruity note higher than 0 and, more essentially, a median of zero defects, thus, having perfect aroma and flavor [12]. EVOO’s flavor and aroma are highly valued properties of this gold ingredient, and they are indicators of quality [13,14]. In addition, the composition of the olives and, therefore, the organoleptic and sensory parameters of the resulting oil can vary greatly depending on intrinsic factors (olive variety, cultivation conditions, etc.) that form a specific phenotypic fingerprint, extrinsic factors (sun exposure, irrigation, production system, storage, packaging, etc.) [3,11,12,15] and the process or the conditions to obtain the EVOO [16].

Most of the volatile compounds present in VOO are synthesized through the interaction between enzymes and substrates during the olive fruit crushing. The lipoxygenase (LOX) pathway participates in the biosynthesis of six straight-chain carbons (C6) compounds, which with their aldehydes, alcohols and corresponding esters, are the most important in VOO aroma [17]. 

The determination of EVOO’s aroma is a complicated matter, since there are numerous variations produced by the mentioned factors, and since possible synergistic or antagonistic effects also come into play [18,19]. During EVOO’s shelf life, these compounds are vulnerable to degradation, mainly due to oxidative processes, which produce alterations in the composition of EVOO [15], and therefore in its organoleptic properties. Volatile pull is actually used as a tool to determine the norms of product acceptability or are even a peculiar indicator of the quality or anomalies due to the presence of substances, which can carry unpleasant smells commonly named “off-flavors” with chemical or microbiological origin [20,21]. 

Due to the knowledge and study of the compounds present in the EVOO’s composition, both degradation alterations and some fraudulent adulterations can be detected [22,23,24]. The identification of the aroma characteristics of VOOs can be carried out by two procedures: sensory assessment (the so-called panel test) and analysis of volatile compounds. The first one has the great disadvantage of being a lengthy and expensive methodology whose result might be affected by many factors such as, for example, the panelists’ training and an inherent subjectivity [25]. 

One of the most suitable analytical techniques for the determination of the volatile fraction is gas chromatography–mass spectrometry (GC–MS). Head-space solid-phase micro-extraction (HS-SPME)-GC-MS is very accurate and precise regarding the qualitative and quantitative study of the volatile fraction of the targeted sample. On the contrary, the preparation of the sample, the analysis and the data management are time consuming, require trained and dedicated lab staff and often are very costly [26,27]. 

For this reason, it is not only important to identify new rapid technologies and devices to support the production of high quality EVOO, but also to improve knowledge regarding the aromas’ evolution, which occurs during the olive oil extraction process and will then be affected during storage [15]. 

However, many alternative techniques have caught on, in recent years, with the objective of displaying faster, easier and cheaper performances. The rapid methods currently available are still considered unsatisfactory, both due to the long analysis time and to the poor reliability and high costs. For many technologies currently in use, the high-speed response often recalls a lack of sensitivity or accuracy. There is, however, another alternative, the use of chemo-sensors or an electronic nose, which has the advantages of low cost and easy sample handling but the disadvantages of poor selectivity, signal drift, and humidity dependence [25].

As a result, interest in new technologies based on chemical sensor arrays [28,29] has grown in recent years. Ample interest has been demonstrated by the numerous scientific publications that are distributed both between classes of foods such as meats, vegetables, cereals, etc., but also between raw materials and packaged products, following the entire production chain from the fields to the fork. 

Applications take into consideration geographical origins, production anomalies, supply chain checks or possible chemical and physical contamination of the matrix. One example could be the use of sensors, and more specifically gas sensors regarding VOCs [30,31,32]. There is a need for the development of accurate instrumental techniques capable of performing measurements in real time and generating the same information as a panel, in a reproducible and stable way, in order to achieve the correct classification of EVOOs rapidly and efficiently. 

The aim of this study was to provide characterization of the aromatic profile of high quality EVOOs, thus allowing for the identification of compounds that mostly contribute to the identification of a particular product using a holistic approach. In summary, this study will characterize the volatile compounds emitted by different oils using GC–MS with SPME analysis and trained panelists from the Slow Food organization. In addition, an innovative technique, based on nanowire gas sensors that can be an advantageous online decision-making aid to the business transformation process, has been applied.

## 2. Materials and Methods

### 2.1. Sample Selection

Ten different EVOOs (Table 1), selected from the Italian Guide of Extra-virgin Olive Oil belonging to Slow Food organization, were analyzed in the present study. All these EVOOs were obtained from olives harvested, processed and bottled during autumn 2020 in Umbria, central Italy.

This little Italian region is placed in the center of the Peninsula, and EVOO production is limited compared to other Italian regions such as Apulia, Calabria, Sicily and Latium, where the quality of EVOO is well known worldwide. Moreover, the analyzed EVOOs have been selected from the above-mentioned official guide for their extreme quality in the 2020 harvest season as confirmed by the awards obtained. Moreover, the Umbrian EVOOs selected are based on the olive cultivar, type of EVOO blend or monovarietal, as well as the geographical location of the olive orchards in order to include all the Umbrian olive cultivation areas. 

The Umbria region has five different PDOs (Protected Denomination of Origin) and three out of ten analyzed EVOOs belonging to different regional denominations: Colli Assisi-Spoleto (OIL-01), Colli del Trasimeno (OIL-05) and Colli Martani (OIL-06). The other seven EVOOs were included: an organic oil produced by high percentage of cv. Frantoio drupes located close to the Trasimeno Lake (OIL-08); six monovarietal EVOOs, one belonging exclusively to cv. Moraiolo (OIL-02) that grew in the so-called “Assisi-Spoleto Olive Belt”; the second was extracted from the minor cultivar Borgiona (OIL-03), a local variety only present in the northeast of the Umbria region; the third selected oil was produced from monumental olive trees belonging to cv. Dolce Agogia (OIL-04), mostly diffused around the Trasimeno Lake and Perugia city; the fourth EVOO from the centennial variety Raio (OIL-07), exclusively grown in the southwest of Umbria; the fifth from another minor cultivar Limona (OIL-09) again from the northeast of the analyzed Italian region; and finally, the last EVOO was obtained from drupes of cv. Frantoio (OIL-10) from southwest of the region.

### 2.2. Experiment Design

The research was carried out following three different methodologies: (i) the determination of volatile organic compound (VOC) of EVOOs through GC–MS and SPME analyses; (ii) organoleptic panel test that was performed by the Taste Panel of the Slow Food organization following the IOC method for the organoleptic assessment of virgin olive oil (COI/T.20/Doc. No 15/Rev. 10 2018) with few modifications; (iii) the application of a Small Sensor System (S3) nanowire gas sensor device that was developed and optimized with collaboration, Nano Sensor System (NASYS) S.r.l. (www.nasys.it, accessed on 19 July 2022), an innovative start-up and spin-off of the University of Brescia.

### 2.3. Sample Preparation

The closed bottles of each EVOO were stored under dark condition in a temperature ranging from 15 to 20 °C for three months (the period from production to commercialization) [33]. In total, 2 mL of each sample was placed in a vial, sealed with septa made of polytetrafluoroethylene (PTFE)/silicone. Once closed, the samples were immediately analyzed in order to prevent the oxidation and the consequent change of the headspace inside the vial. Samples used were treated and prepared for GC–MS SPME and S3 techniques, exactly in the same way, to reduce the variables related to the preparation stage as much as possible. The operational conditions are fully reported in the sections below as well as for data analysis.

### 2.4. GC-MS SPME Detection

Vials were placed in the auto-sampler HT280T (HTA s.r.l., Brescia, Italy) to proceed with vial conditioning and volatile organic compound (VOC) extraction. Conditioning of the sample was performed as follows: filled vials were maintained for 5 min at 40 °C in order to equilibrate the headspace (HS) of the sample and to remove any variables. Afterward, VOC extraction was performed using SPME analysis, and the fiber used for the adsorption of volatiles was a divinylbenzene/carboxen/polydimethylsiloxane (DVB/CAR/PDMS) 50/30 µm (Supelco Co., Bellefonte, PA, USA) placed on the HT280T auto-sampler. The fiber was exposed to the vial HS in the HT280T oven thermostatically regulated at 40 °C for 20 min. 

The GC instrument used in this work was a Shimadzu GC 2010 PLUS (Kyoto, KYT, Japan), equipped with a Shimadzu single quadrupole mass spectrometer (MS) MS-QP2020 (Kyoto, KYT, Japan). Fiber desorption took place in the GC–MS injector for 6 min at 220 °C. GC was operated in the direct mode throughout the run, while the separation was performed on a MEGA-5MS capillary column, 25 m × 0.25 mm × 0.25 µm, (Agilent Technologies, Santa Clara, CA, USA). 

Hydrogen was used as the carrier gas and has been produced by GENius PF500, FullTech Instruments Srl (Rome, Italy) at a constant flow rate of 2.00 mL/min. The GC oven temperature programming was applied as follows: at the beginning, the chromatographic column was held at 40 °C for 5 min and, subsequently, the temperature was raised from 40 to 190 °C at 4 °C/min and held for 1 min for a total program time of 43.50 min [24,26,27]. During the analysis, the GC–MS interface was kept at 220 °C, with the mass spectrometer in the electron ionization (EI) mode (70 eV) and related to instrument tuning, and the ion source was kept at 200 °C. Mass spectra were collected over 35 to 500 m/z in range in the total ion current (TIC) mode, with scan intervals at 0.2 s. VOC identification was carried out using the NIST11, NIST11S and the FFNSC2 libraries of mass spectra. 

Chromatogram peak integration was performed using the peak area as target parameter programming, an automatic integration round using 70 as the minimum number of peak detection, and 500 as the minimum area to detect. Other parameters used in the automatic peak integration were: slope 100/min, width 2 s, drift 0/min, doubling time (T.DBL) 1000 min, and no smoothing method was applied. The final round of peak integration was performed by manual peak integration for all the obtained chromatograms.

### 2.5. Organoleptic Analysis

The organoleptic profile of the oil samples was evaluated by the Taste Panel of the Slow Food organization (https://www.slowfood.it/grande-olio-2021-guida-ai-riconoscimenti-regione-per-regione/, accessed on 19 July 2022), following the IOC method for the organoleptic assessment of virgin olive oil (COI/T.20/Doc. No 15/Rev. 10 2018) with slight modifications (Mousavi et al., 2021). The most important differences concerned the number of tasters, which was five plus a leader while all the other recommendations were applied. 

The presentation of each sample to the panelist was blind. The tasting session begins with a calibration between the members, and the panel leader provided different oils and evaluated the response of each member. After the calibration, the panel leader, in a separate room, numbered every replica of oil and then offered to the panel members one glass with the same code at a time with a maximum of three oils for each section. 

Moreover, the panel leader checked the results, performing the average of them, and controlled if reported results were out of range. If this last case occurred, the panel leader started a new calibration test, and the sample was numbered and tasted again. Olfactory sensations were evaluated considering positive attributes, i.e., fruitiness and persistence, and, eventually, negative traits, as the presence of defects (i.e., fusty/muddy sediment, musty/humid/earthy, winey/vinegary/acid/sour). 

The panel leader compiled the notes given by each taster, and the statistical evaluation was carried out by the median of each parameter. This test provided sequential information about the sensory characteristics of the samples and identified the organoleptic profile of EVOOs belonging to the different companies and cultivars along the Umbria region. Eight out of ten analyzed EVOOs from the Umbria region have been also tested by seventeen Slow Food panel leaders from all Italian regions plus the head of leaders in the final selection of high-quality EVOOs (https://www.slowfood.it/grande-olio-2021-guida-ai-riconoscimenti-regione-per-regione/, accessed on 19 July 2022). The age of the panelist was from 30–60 years old, the majority of them were males, and all had the several years of experience in the olive and oil sector and the organoleptic panel test experience at national and international scales.

### 2.6. S3 Detection

The autosampler used for the S3 device is the same model also used to prepare the samples seen in Section 2.4 associated with the GC–MS. S3 is equipped with an array of chemiresistor-type sensors, and the conditioning of the samples was the same as well. The VOCs collected by the auto sampler is conveyed inside the sensor chamber due to the pneumatic pump. A total number of 20 replicates for each sample for a total of 200 replicas were carried out. 

The Small Sensor System (S3) is a device that has amply demonstrated the advantages of the application of this technology in recent years. This technology is based on semiconductor metal oxide gas sensors (MOX) [34,35], used with considerable success in several sectors, ranging from food safety [36], quality control [37,38,39], environmental monitoring and human health, in particular due to its high sensitivity, fast responses and low costs [40]. The innovative S3 device consists of an array, made of different semiconductor metal oxide gas sensors, flow, temperature and humidity sensors. 

The metal oxide sensors mechanism of operation is based on the variation of the electrical conductance of the sensing material, a deposition of a semiconductor metal oxide, caused by interaction with the gaseous surrounding environment. The reaction between the oxygen species adsorbed on the surface of the sensitive element and the target molecules present in the gas samples causes a release of electrons, which in turn modulates electrical properties, including electrical conductance and resistance [41]. 

The surface of the thin film is rough, and this is an advantage since it provides a high surface-to-volume ratio and reactivity with gaseous species [42]. In addition, the existence of such very rough surface morphology gives rise to a high specific area required for high-sensitivity gas sensors [43]. S3 is composed of three essential parts:

1. Sensor’s chamber: three MOX sensors (Table 2) are positioned into a steel chamber separated from the external environment, except for an inlet and an outlet path for the passage of volatile compounds. In addition to the MOX sensors, there are also temperature, humidity, and flow sensors, which are fundamental to take into account the number of features during the process. The dimensions of the chamber are 11 × 6.5 × 1.3 cm.

2. Fluid dynamic circuit for the distribution of volatile compounds: the fluid dynamic circuit is formed by a pump (Knf, model: NMP05B), polyurethane pipes, a solenoid valve, and a metal cylinder where there is an activated carbon for filtering any type of odors present outside of the instrument. The pump flow is set by a needle valve positioned at the chamber inlet.

3. Electronics control system: the electronic boards can register the resistance variations of the sensors, their correct heating, their operating temperature, and permit as well to send the registered data in real time to the dedicated Web App through an internet connection that can store and analyze the data acquired in the cloud making S3 an IoT device [31]. In addition, it allows for communication and synchronization with an auto sampler.

### 2.7. Data Analysis

From GC–MS analysis, a list of all the VOCs of the samples was constructed in order to identify which compounds were in common for different samples. Common VOCs were obtained comparing VOC lists of GC–MS analysis, considering all the different samples. Only VOCs present in at least 2 of the 3 replicates of all samples were considered. Afterward, compounds were classified based on the chemical group families, and the percentage of each group over the total was calculated. 

The organoleptic results were summarized in ten different spider graphs, which include all the olfactory perceptions individuated in the analyzed EVOOs in a numerical scale from ‘0’ (not found) to ‘5’ (highly individuated). The data obtained from the S3 device were developed using principal component analysis (PCA), which was performed using GraphPadPrism 9 software for volatile data, organoleptic results and S3 detections (performing a mean of all lectures for each sensor used), both to understand the ability of these techniques to separate all analyzed EVOOs and to see the different separations in the hyperplane of our samples after the application of each method. This technique consists of clustering the sample variables, through linear combinations, that describe the link between one sample to the others, obtaining the principal components (PC), which are far fewer than the original variables. PCA can reveal all possible clusters of samples linked by similar characteristics within the main components considered in the hyperplane. Data analysis was performed using GraphPadPrism 9 software. Sensor responses in terms of resistance (Ω) were normalized when compared to the first value of the acquisition (R0). For all the sensors, the difference between the first value and the minimum value during the analysis time was calculated; hence, ΔR/R0 has been extracted as a feature.

To find any correlation among the three methodologies and the correlations inside the data of each method, Pearson’s correlation coefficient was performed by GraphPadPrism-9 software. The *p* value under 0.05 was taken into consideration, and the correlations ranged between −1 and 1 were discussed.

## 3. Results and Discussion

### 3.1. GC-MS SPME Detection

The ten analyzed EVOOs could individuate 71 different compounds, which were found at least in two out of three different detections. In particular, ten different classes of compounds were detected: alkanes, carboxylic acids, alkenes, alcohols, aldehydes, ketones, dienes, diols, esters and terpenes. (Table 3).

Within the 71 compounds, only the hexanal and 2-hexenal, both from the aldehyde category, were present in all analyzed EVOOs and at a high percentage of abundance (Table 3). This result was in accordance with previous studies on VOCs in different EVOOs, in which the hexanal was one of the major VOCs [33,78,79]. Hexanal gives a great contribution in the formation of the majority of green attributes [80,81]. García-Vico et al. in 2017 [11] reported the highest percentage of 2-hexenal with respect to the other C6 VOCs. C6 aldehydes, alcohols, and their corresponding esters are considered, both qualitatively and quantitatively, the most crucial and influential aroma compounds of EVOO. They are related to sweetness and green notes and contribute favorably to the aroma [11,12,13].

In another study, the highest concentration of 2-hexenal in all analyzed oil samples has been reported [33]. EVOOs with a maximum abundance of 2-hexenal were OIL-01 and 10, which in both of them the cultivar Frantoio was present, especially for the second one, which was a monocultivar. This result was in accordance with previously published studies [81,82,83] who reported the highest percentage of 2-hexenal in Frantoio oil. Several research studies reported the maximum presence of 2-hexenal with respect to other VOCs, while in the present study, hexanal and 2-hexenal were present mostly with the same abundance in all the analyzed EVOOs. Moreover, among these, hexanal had the highest percentage of abundance in OIL-02, 03, 08 and 09. On the contrary, 47 out of 71 detected compounds were private to a single EVOO. Concerning the alkane category, 1-Iodo-2-methylundecane was detected only in OIL-03, while there was 2-isothiocyanato-butane in OIL-07, which was a unique oil with the 2,2,4,6,6-pentamethyl volatile compound. Furthermore, 2,6,7-trimethyl-decane, 4,6-dimethyl-dodecane and heptane were only present in OIL-02. This result was in accordance with Blasi et al. [82] for the high percentage of n-decane in Moraiolo oil with respect to Leccino and Frantoio EVOOs. The GC–MS SPME results within the carboxylic acid category detected two individuated compounds, 1,7,7-trimethyl-bicyclo[2.2.1]hept-2-yl ester, which was found only in the cv. Dolce Agogia EVOO (OIL-04), and acetic acid, hexyl ester detected in eight out of ten EVOOs excluding monovarietal oils from cvs. Raio and Limona (OIL-07 and OIL-09, respectively) [84]. Four chemical categories (alkenes, diols, esters and terpenes) had a main role in the discrimination of EVOOs. The average number of detected compounds among all analyzed EVOOs was sixteen. Noteworthy, OIL-02, and OIL-04 had the highest number of detected VOCs, 20 and 31, respectively. Furthermore, the EVOO coded by OIL-04 had the highest number of terpenes (nine) when compared with all the other analyzed oils (Figure 1 and Table 3). OIL-04 was the most complex one for the presence of VOCs from different categories, while the EVOO with almost balanced abundance of VOCs was OIL-07 from Raio cultivar [85,86].

The PCA analysis performed in the ten EVOOs (Figure 2) was able to separate clearly six out of ten olive oils. In particular, OIL-03, 09 and 10 were placed in the positive plot area. The biplot graph shows that the VOCs that participated in this placement were 3-methyl-1-butanol, 3-pentanone and hexanal. The EVOOs with highest abundance of 3-methyl-1-butanol were OIL-03 and OIL-09, which are from local Umbrian cultivars growing in the same area of the region. This VOC was reported to be responsible for sweet sensorial sensation in high quality olive oils [78]. 3-pentenone had the highest abundance in OIL-09, which is responsible for fruity, green and sweet sensation [79].

OIL-04 was extremely distant to the others, occupying the negative plot area. The biplot graph shows that the presence of α-pinene was responsible for this division. The most abundant VOC in OIL-04 was α-pinene (28.98%), which is responsible for pine aroma perception [51]. The presence of this VOC was reported in other cultivars such as Leccino, Coratina and Intosso in very low concentrations [78]. Even though the presence of α-pinene and copaene could be related to the pedoclimatic condition of the olive tree, especially the altitude [81], in the present study, OIL-04 and 05 were from the same geographic condition and were extracted by the same olive mill. This abundance of α-pinene present only in OIL-04 and in a very low percentage in OIL-05 could be mostly related to the cultivar stated by the producer, Dolce Agogia, allowing to hypothesize a genotype effect. A previous study has highlighted the minor effect of the climate variable and the dominant effect of cultivar on the formation of the oil aroma [80]. The EVOO 07 obtained from an Umbrian local variety in the south of the region was located at the positive plot area for PC 1 and negative for PC 2. This EVOO was the unique one to have 2-methyl-1-propanol previously reported as responsible for the green perception of olive oil [25].

### 3.2. Panel Test

The Slow Food panel test revealed different olfactory perceptions among the ten analyzed EVOOs. The panel revealed any defect, while nine different attributes in total were reported: fruity, flavor, cinnamon, artichoke, tomato, balsamic resin, herbal, wild flowers and almond. The spider graphs include all the sensorial perceptions detected by the tasters (Figure 3).

The fruity perception was detected, at different amounts, in all EVOOs, while “wild flowers” characterized the EVOOs coded as OIL-. This could be due to a typical blend of different olive cultivars that when used as monovarietal, as in the case of OIL-04 and OIL-10, did not give the same perception. In a previous study, it has been reported that the percentage of fruit from each cultivar has influence on the quality and quantity of VOCs [80]. “Cinnamon” was detected only in OIL-; the peculiarity did not depend on the pedoclimatic conditions or on the extraction technology, which are the same as for OIL-02. For this reason, we could hypothesize that the cinnamon perception derived from a specific blend among different olive varieties. The “tomato” smell was detected in only two monovarietal EVOOs from cv. Borgiona and the monovarietal of cv. Raio. These two EVOOs were produced under different pedological and climatic conditions such as temperature and humidity; the tomato smell that is not common in the Umbrian cultivars could be under genotypic control. The “almond” was detected in three out of ten EVOOs (OIL-06, OIL-08, OIL-10), which belong to a very different place, but all of them were produced with a high percentage of olive fruit from cv. Frantoio. The “balsamic resin” perception was individuated in only two EVOOs, OIL-04 and OIL-05, which were correlated by the presence of several cultivars to constitute the blend and by a presence of two ancient Umbrian local varieties as Dolce Agogia and San Felice, respectively. Moreover, all these EVOOs demonstrated a complex olfactory perception formed mostly by three perceptions at maximum level. The EVOOs with more than three effective perceptions were OIL-07, OIL-08 and OIL-10, of which two of them belong to the monovarietal oil of cv. Raio and cv. Frantoio from the southern Umbria region. The presence of four perceptions, out of nine, highlighted how even in a monovarietal olive oil the bouquet could be complex (Figure 3). The PCA analysis, with explained variance (EV) equal to 60.7 and expressed by 36.20% for PC1 and 22.50% for PC2, performed on the ten EVOOs by using the data from the organoleptic test, was able to clearly separate eight out of ten EVOOs (Figure 4). The extra virgin olive oils OIL-02 and OIL-03 were placed close to each other in the positive plot area. The biplot graph shows that this placement was the cause of high flavor, fruity, artichoke and tomato perceptions in these two EVOOs. All the other oils are distributed far from each other, as also observed in the PCA of volatile compounds. OIL-08, 09 and 10 were positioned in the negative plot area, since they had a high perception of herbal and wild flowers in the case of OIL-08. EVOO 07 was the unique one to be placed in the positive area for PC 1 and in the negative area for PC 2, of which the biplot graph shows this division that is because both the tomato and artichoke perceptions are almost at the same level in this oil. The other four EVOOs were placed near to each other for their cinnamon, balsamic resin and almond perceptions (Figure 4). The EVOOs with the same position in the PCA plot area of VOCs and panel test results (Figure 2 and Figure 4) were OIL-03, which was in the positive plot area in both PCA analyses, as well as OIL-01, OIL-06 and OIL-07 sharing the same position. Comparing these PCA results, in OIL-03, the peculiarity was a high perception of tomato with respect to the other EVOOs. Except for VOCs from alcohol and aldehyde categories, which were present in almost all EVOOs, for tomato perception, the responsible VOCs, which were only present in this EVOO, were hexadecane, 2,4,6-trimethyl-octane, 4-methyl-1-heptene, (Z)-9-methyl-5-undecene, n-hexanol and a low amount of copaene. In previous studies, the tomato perception was related to Z-3-hexenal, Z-3-hexenol, and Z-3-hexenyl acetate [87,88]. In another study, except for alcohols and aldehydes, copaene was also related to the tomato and artichoke perception [89]. The peculiarity of OIL-06 was the almond perception with respect to the other EVOOs. The VOCs, which could be responsible for this perception, were hexane and 4-methyl-decane, with the highest abundance in this oil and 2-methyl-nonane, which was only present in this EVOO. Almond flavor has been recurrently associated with (Z)-2-penten-1-ol, as well as hexanal [18]. In EVOO 06, more than the above-mentioned VOCs hexanol, hexanal, 2-hexenal and 3-pentanone were also present. The unique EVOO with cinnamon perception, OIL-01, in addition to the high abundance of hexanol, hexanal, 2-hexenal and 3-pentanone, which could be related to the fruity and flavor perception of this EVOO, had some VOCs private to this oil. (E)-11-tetradecen-1-ol, (Z)-2-penten-1-ol, 3,7-dimethyl-6-octen-1-yn-3-ol and (E)-4,8-dimethyl-1,3,7-nonatriene were only present in OIL-01. Finally, EVOO 07, which was positioned in the same plot area of both PCA analyses (Figure 2 and Figure 4), had the artichoke, tomato and herbal perception together. 4,8-dimethyl-1,7-nonadiene was present only in this oil and OIL-02 in which artichoke had a high abundance. In previous studies, the artichoke perception was correlated positively to 1-penten-3-one, trans-2-pentenal and 1-penten-3-ol,1-hexanol, 2-hexen-1-ol, 3-hexen-1-ol and 2,4-hexadienal [87,90], some of which were present in EVOO 07.

### 3.3. S3 Detection

Regarding the results obtained with the S3 (Figure 5), it was observed that, in general terms, all 20 replicates of each EVOO clustered well in the hyperplane space, with few exceptions demonstrating an intra-sample variance. A total number of 200 observations, belonging to an average of 20,000 lectures of the three applied sensors, have been registered. It was remarkable how 97.10% of EV (explained variance) determined by PC1 and 2.42% by PC2 were enclosed in the hyperplane.

The twenty replicates of EVOO 03 and almost all of the EVOO 02 were unique in that they placed in the negative area for PC 1 and positive area for PC 2; they are also the EVOOs where the 20 observations are considerably scattered in the hyperplane. The proximity of the 20 points of these two EVOOs is possibly explained by similar values in terms of aldehydes, ketones and alcohols. EVOO 01 was placed in the negative part of both axes of PCA, together with EVOO 07. These two olive oils had similar percentages of hexanal, 1-hexanol and 3-pentanone, as well as two dienes as observed in the VOCs analysis. EVOO 08, together with OIL-09 and OIL-10 were placed in the positive plot area of PC1 and negative area of PC2, while all 20 lectures of EVOOs 04, 05 and 06 were placed in the positive plot area. EVOO 04 in the positive plot area represented optimal clustering; it had a complex and fragmented aromatic profile showing a high percentage of terpene α-pinene. OIL-04 was the unique one with a high perception of balsamic resin; a high abundance of α-pinene in this EVOO, which is responsible for pine aroma [85], could be responsible for resin perception. The proximity of EVOOs 05 and 06 in the same plot area could probably be derived from the genotypic effect of these two oils, which are both the blend of cultivars Frantoio, Moraiolo and Leccino with some percentage of other cultivars and with almost the same level of fruity and flavor perception. The proximity of EVOOs in the PCA area was confirmed by the data obtained with the GC–MS SPME detection. The EVOOs with a very similar volatile profile; in particular, high percentage of the volatiles such as 2-hexenal, 1-hexanol and 3-pentanone, were placed near to each other in the plot area. EVOO 09 showed a volatile profile similar to the 08 and 10, from which it differed for the remarkable presence of 3-methyl-1-butanol, which represented one of the compounds with a higher concentration in this olive oil. Another PCA analysis (Figure 6) was performed by using the mean value of each sensor observation in each EVOO in order to better indicate the potentiality of these sensors to clusterize the ten olive oils. EVOOs 01, 02, 03 were completely separated in the hyperplane and are far from the other EVOOs. This result, even if not in perfect accordance with the VOCs and organoleptic ones, helped to clarify the differentiation of some EVOOs that clustered together when the other two techniques were applied.

There is a growing emphasis on sensor arrays or electronic noses able to give information about quality control and classification of EVOOs from different geographical areas [91,92]. In recent years, electronic nose systems with metal oxide semiconductor gas sensors have received much attention in the literature for the determination or classification of the geographical proveniences of EVOOs [93,94]. Changes in resistances values of sensors when the surface of the sensing element comes into contact with a pull of volatile compounds can tell us different things about the quality of the products, extraction conditions, origin and conservation method [31,95]. The analyzed EVOOs come from a little geographical area and belong to a very high-quality level of olive oils. Taking this into consideration, the sensor results, even if they could be optimized, are more than promising to individuate the peculiarity of each studied EVOO and their classification. Due to the operation features of the sensor device, which provide an answer in less than a minute once the database is trained, could be a support to all the classical chemical analyses commonly performed in the EVOOs production chain that are time consuming and need specifically trained lab staff, as well as a specific panel with a numerically limited capacity of analysis. Furthermore, the application of the sensors will reduce the cost of the analysis performances, since no reagents are necessary for the analysis nor for a lab facility. In addition, sensor arrays, as aforementioned, have the possibility to be customized in a hand-held device with a user-friendly interface in order to be used by anyone with no scientific background directly on the fields or forming an array network in the production chains in order to monitor in real time the production process.

### 3.4. Pearson’s Correlation among the Three Different Methodologies

All three methodologies applied to differentiate ten of the best Umbria EVOOs of the harvesting season 2020–2021 have achieved excellent results. The correlations among sensory perceptions detected by panelists or by electronic sensors with the volatile compounds were deeply investigated. The goal of this research was to simplify the subdivision of EVOOs by a sensor, which should summarize all the volatile compounds detected by GC–MS SPME and the olfactory perceptions of professional panelists in a few seconds. To achieve this result, we started from a homogeneous olive oil category, which was recognized with an extreme high quality by the Slow Food official guide in 2021, and which came from a little Italian region where olive oil production, even with a very high-quality level, is limited to a few tons per year. In the previous paragraphs, it was reported how each method had a very interesting potentiality to discriminate high-quality EVOOs, but in order to better individuate the relationship among the three applied methods, a Pearson’s correlation was performed (Figure 7 and Appendix A).

The “r” results together with the *p* value from each correlation between two factors summarized the relationships within the same category (volatile, organoleptic and sensor), and between the three categories, the range went from a maximum of positive correlation equal to 1 to a maximum of negative correlation equal to −1. Furthermore, 388 correlations (*p* < 0.05) were found through Pearson’s analysis, 45 negatives and 343 positives. Among them, 307 were found within the volatile compounds, 3 among sensors (equal to 100%) and only 1 within the organoleptic parameters. In addition, 77 correlations among volatiles, sensors and organoleptic analyses were found, including: 17 between VOCs and sensors in which 2 alkanes were positive and 1, undecane, was in negative correlation with the sensors; 54 correlations between VOCs and organoleptic; and finally, 6 correlations among sensors and organoleptic data (Figure 3 and Appendix A). Alkanes, terpenes, alcohols and dienes were the categories with the highest correlation to the organoleptic perceptions. Among VOCs correlated to organoleptic perceptions, only two of them, decane and hexane, had positive correlation with fruity perception (r ≥ 0.68). This perception was mostly presented in OIL-02, 05 and 06; noteworthy in all of three EVOOs, the cv. Moraiolo was present. Three VOCs were correlated with flavor, undecane and tridecane were negatively correlated (r ≤ −0.69), while acetic acid hexyl ester was positively correlated (r ≥ 0.70). The organoleptic test indicated that OIL-03, 05 and 06 had the maximum level of flavor. OIL-01 was the only one with the cinnamon perception detected from the panelists. Six VOCs were correlated with cinnamon taste, and among them, three alcohols showed maximum and positive correlations (r = 1), (E)-11-tetradecen-1-ol, (Z)-2-penten-1-ol, and 3,7-dimethyl-6-octen-1-yn-3-ol, as well as one VOC of dienes categories, (E)-4,8-dimethyl-1,3,7-nonatriene. Six VOCs were correlated with artichoke sensation perceived in OIL-02 and 07 during the organoleptic analysis, all of them with r ≥ 0.80. Eight VOCs were correlated to tomato taste (r ≥ 0.79), and among them, only one, (E)-3-Hexen-1-ol acetate with a negative correlation (r ≤ −0.74). OIL-03 of cv. Borgiona and OIL-07 of cv. Raio were the only ones with this perception. Twenty VOCs were correlated with balsamic resin sensation, and almost all of them were positively correlated (r ≥ 0.81), except for hexanol, which was correlated negatively (r = −0.68). In other studies, hexanal has been reported to be negatively correlated with leaf and lawn attributes, while it was positively correlated to almond attribute [33], the balsamic resin perceived could be added as a new perception that increased when the amount of hexanal decreased. The EVOO with a maximum level of this perception was OIL-04, and a lesser amount in OIL-05. In both of these EVOOs, as reported above, the cv. Dolce Agogia was present especially in OIL-04. Five VOCs were correlated to herbal sensation, four were positively correlated (r ≥ 0.65) and one was in negative correlation, acetic acid hexyl ester (r ≤ −0.89). The maximum level of this sensation was present in OIL-09. Two VOCs were correlated positively with wild flowers; both of them from alkanes, and the category was only present in OIL-08. Almond sensation had positive correlation (r ≥ 0.75) with two VOCs, 4-methyl-decane and 2-methyl-nonane, both from the alkane category with the highest level in OIL-06 (Figure 3 and Appendix A). The three sensors were positively correlated, and among them ΔR ≥ 0.90. A correlation was also found between the sensors and 17 of the whole pull of VOCs. *p* < 0.05 was selected as the minimum value to determine whether or not the correlation was significant, and all the correlations that passed showed values of ΔR −/+ 0.66. A positive correlation means that increasing the concentration of the specific volatile compound increases the ΔR of the sensor as well. This phenomenon has been observed in eight compounds (three alkanes, one ester, two alcohols, one aldehyde and one ketone) with their characteristic scent 2,6,7-trimethyl decane (acetic, moldy) [52], dodecane (dry musk, essential oil of zingiber officinale) [96], 4,6-dimethyl- dodecane (floral, present in sweet violet Viola odorata) [52], acetic acid, hexyl ester (fruity, green, apple, banana, sweet) [97], 1-pentanol (fusel, fermented bready, cereal and fruity) [53], 3,7-dimethyl-3-octanol (clean, fresh, floral tea, citrus and herbal) [53], nonanal (aldehydic, citrus, cucumber, melon, rindy potato) [97] 5-ethyl-4-methyl 3-heptanone, (fruity, citrus) [53]. On the contrary, a negative correlation means that with the increase in the concentration for the different compounds, a decrease in the ΔR of the sensor response is observed. This behavior was observed in two compounds, 3-pentanone (fruity, green, sweet) [98,99] and undecane (herbal eucalyptus woody thujonic) [53]. Regarding the correlations between the sensor response and the organoleptic parameters, three positive correlations for flavor and three negatives for herbal perceptions were detected (Figure 3 and Appendix A). For all three sensors, by increasing the flavor of the oil matrix, the response of the sensor increased, making the use of the sensor a good tool to trace the intensity of the oil flavor. It was observed as well that all sensors were negatively correlated with herbal perception. These two EVOO characteristics were reported by all the official panel testers as a strong indication of their quality, and then, the S3 here applied seemed to statistically significantly clusterize these two perceptions, and therefore, their application in a large set of olive oils including different quality categories is desirable. The positive correlations among the three methodologies were found for artichoke perception and in detail, with two alkanes, two alcohols and one ketone. Moreover, flavor perception was positively correlated with the sensor as reported above and negatively with undecane, corresponding to herbal perception. As confirmation to what was previously reported, the same alkane was positively correlated with sensor and herbal odor. Finally, the acetic acid hexyl ester was positively correlated with sensor results and flavor smell negatively correlated with herbal perception. Several analytical methods have been proposed in different studies carried out in the last decade, many including characterizations of the volatile markers for each sensory characteristics [100], and there have been recent efforts devoted to the quality testing of different types of oils using electronic sensors [31,32,36,37,38].

## 4. Conclusions

Quality EVOOs stand out among any other vegetal oils due to their unique aroma and flavor characteristics that are intrinsically related to VOCs. This study allowed for a better understanding about the VOCs profile, considering that SPME GC/MS identified more than 70 compounds from different chemical functional groups, mainly aldehydes, short-chain alcohols, ketones and alkanes. These compounds were related to the composition of EVOOs and type of fruitiness characteristics in the samples and were observed to have significant correlations with the sensory profile of the EVOO panel test. The development of easy-to-use sensor devices will have a triple innovative aspect of EVOO production enhancements. First, it will be capable of being installed directly in the process line of EVOO production or will monitor the final product to detect the quality level. Second, it will be a user-friendly support for the producers and farmers since it will be able to create a network and a long-term database directly on the fields and prediction in the selected critical control points. Last but not the least, S3 is a stand-alone device that is able to work in continuous operation mode, providing a response in less than a minute once trained, without the necessity of sample treatment or problems correlated to the number of replicates, appearing extremely appealing in this field.

## Figures and Tables

**Figure 1 sensors-22-07164-f001:**
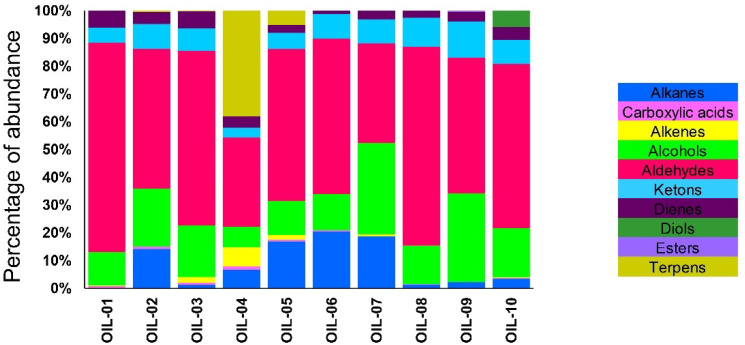
The variation of VOCs in each analyzed EVOO based on the percentage of abundance. VOC category indicated by different color.

**Figure 2 sensors-22-07164-f002:**
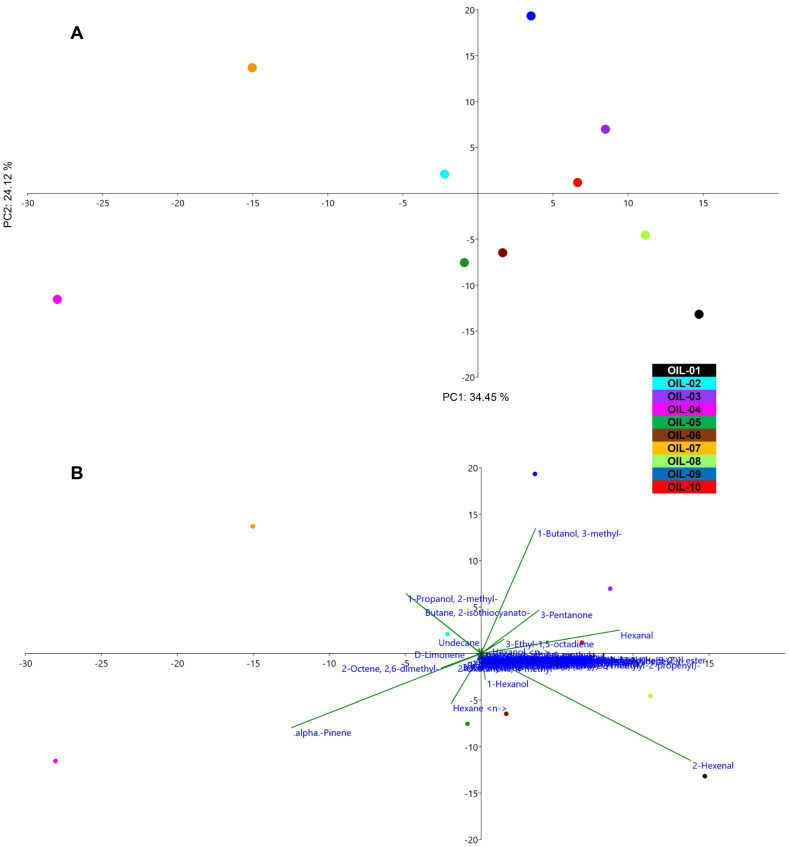
PCA (**A**) and biplot (**B**) representing the distribution of the EVOOs based on the VOC abundance in each oil. EVOOs are indicated by different color.

**Figure 3 sensors-22-07164-f003:**
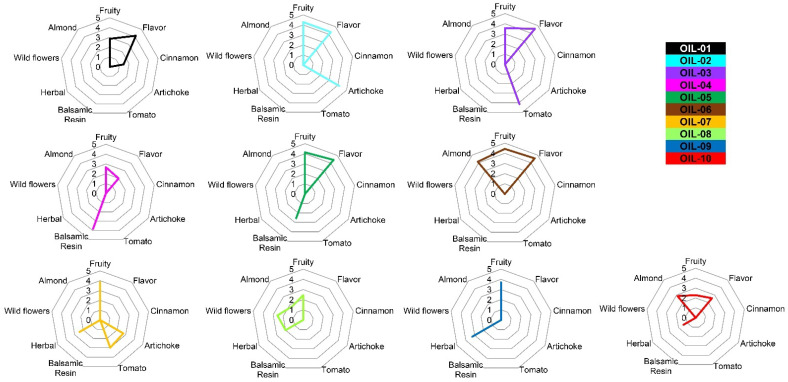
T Radar chart reporting the median of fifteen organoleptic evaluations (three replicas × oil × panel members) carried out on EVOOs. EVOOs are indicated by different color.

**Figure 4 sensors-22-07164-f004:**
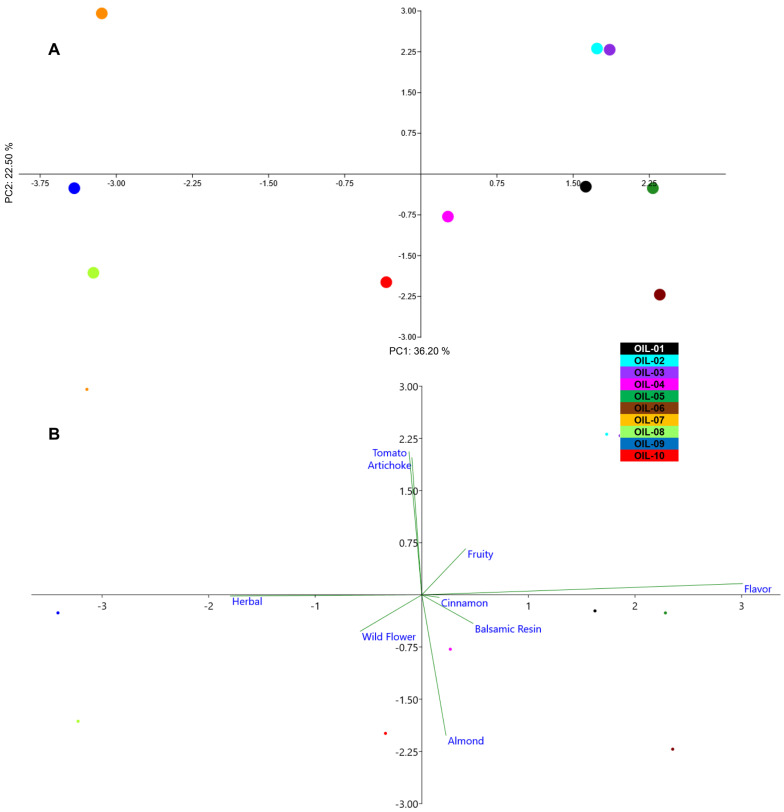
PCA (**A**) and biplot (**B**) showing the distribution of EVOOs for their organoleptic profile. EVOOs are indicated by different color.

**Figure 5 sensors-22-07164-f005:**
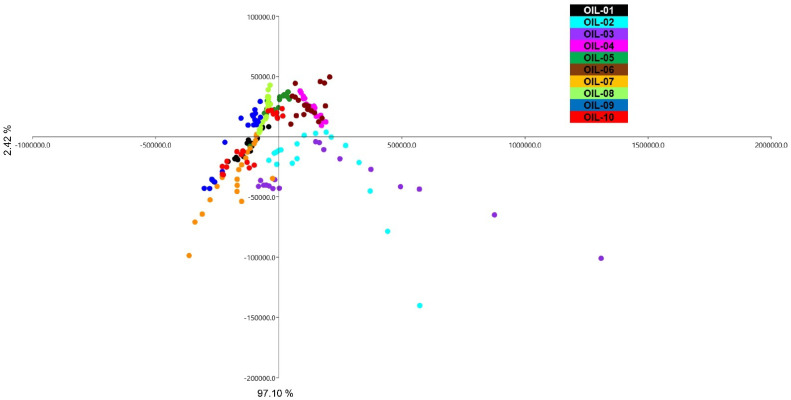
PCA graph representing the distribution of ten studied EVOOs based on the S3 sensors data. Each EVOO is indicated by 20 replicates and different color.

**Figure 6 sensors-22-07164-f006:**
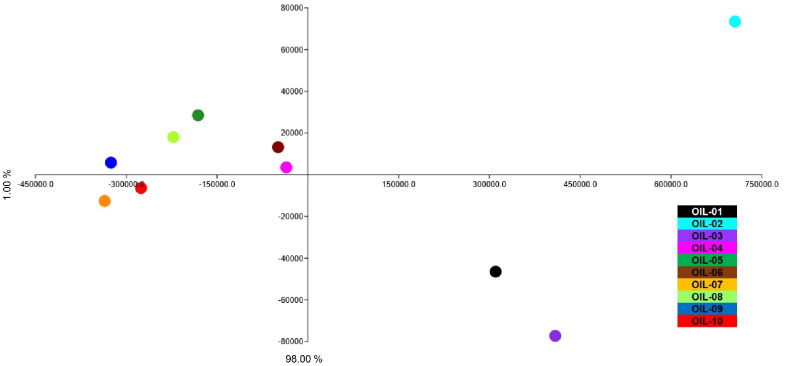
PCA showing mean value of each sensor observation in each EVOO. Colors indicates different EVOOs.

**Figure 7 sensors-22-07164-f007:**
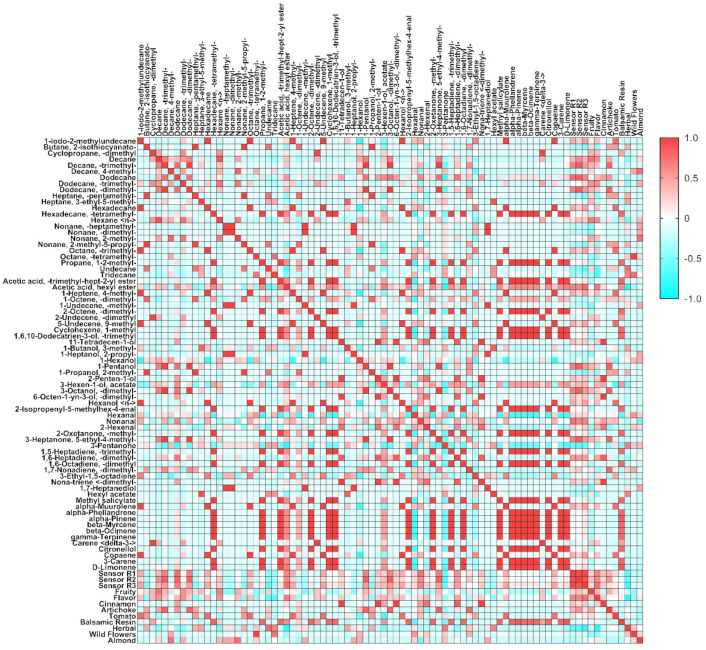
Heat map representing Pearson’s correlation coefficients among the three methodologies applied. The correlation within and between the three different approaches ranged from −1 (**light blue**) to 1 (**red**).

**Table 1 sensors-22-07164-t001:** List of EVOOs analyzed in the present study together with the cultivars (olive genotypes) collected to produce each olive oil. The code assigned to each EVOO, as presented in all elaborations, is reported.

EVOO	Olive Cultivar	EVOO Code
Marfuga—Riserva DOP Assisi Spoleto Bio	Moraiolo, Frantoio, Leccino	OIL-01
Marfuga—L’Affiorante Bio	Moraiolo	OIL-02
Le Pietraie—Borgiona Bio	Borgiona	OIL-03
CM Centumbrie—Dolce Agogia	Dolce Agogia	OIL-04
CM Centumbrie—DOP Colli Del Trasimeno	Moraiolo, Frantoio, Leccino, Dolce Agogia	OIL-05
Decimi—DOP Colli Martani	Moraiolo, Frantoio, Leccino, San Felice	OIL-06
Oliveto di Geltrude Contessa—Rajo	Raio	OIL-07
Fontanaro—Olio della Pace Bio	Frantoio, Dolce Agogia, Leccino	OIL-08
Silvano Di Murro—Limona	Limona	OIL-09
Alessandro Ricci—Frantoio	Frantoio	OIL-10

**Table 2 sensors-22-07164-t002:** Description of sensor array.

Name	Material	Temperature (°C)	Sensor Kind
Sensor R1	SnO_2_	300	RGTO
Sensor R2	SnO_2_Au	400	RGTO
Sensor R3	SnO_2_	400	RGTO

**Table 3 sensors-22-07164-t003:** Volatile compounds detected by GC–MS SPME for each EVOO and divided by chemical category. Each compound is presented in terms of abundance (a dimensional variable) represented the average of three biological replicas and their standard deviation.

Volatile Compound	EVOO Code										Description
	OIL-01	OIL-02	OIL-03	OIL-04	OIL-05	OIL-06	OIL-07	OIL-08	OIL-09	OIL-10	
Alkane											
1-Iodo-2-methylundecane	nd *	nd	0.04 ± 0.07	nd	nd	nd	nd	nd	nd	nd	Linear molecule of 11 carbons substituted by an iodo group at position 1 and a methyl group at position 2. Is a natural product found in Vitis vinifera, identified in *Hypericum mysorense* bark by GC-MS analysis. It has been identified as one of the most prevailing compounds positively correlated with the presence of squalene with antimicrobial activity as well with a non-define scent found in olive oil [44].
Butane, 2-isothiocyanato-	nd	nd	nd	nd	nd	nd	16.48 ± 0.02	nd	nd	nd	Linear molecule of 4 carbons with a substituent in position 2. Is a natural product found in *Brassica juncea*, *Brassica rapa*, and *Eutrema japonicum.* Normally are thioglycoside and glucosinolate degradation products responsible for pungent and green aroma found in olive oil [45].
Cyclopropane, 1,1-dimethyl-2-(1-methyl-2-propenyl)-	nd	nd	nd	nd	2.82 ± 0.44	0.45 ± 0.05	nd	nd	nd	nd	Cyclic molecule of 3 carbons, with a branched substituent in position 1 with a non-define scent found in olive oil [46].
Decane	nd	0.88 ± 0.04	nd	nd	0.61 ± 0.14	0.68	nd	nd	nd	nd	Linear molecule of 10 carbons with a non-define scent found in olive oil [46].
Decane, 2,6,7-trimethyl-	nd	1.97 ± 0.08	nd	nd	nd	nd	nd	nd	nd	nd	Linear branched molecule consisting of decane bearing three methyl substituents at positions 2, 6 and 7 with a non-define scent found in olive oil [46].
Decane, 4-methyl-	nd	nd	nd	nd	nd	1.07 ± 0.58	0.26 ± 0.94	0.68 ± 0.23	nd	nd	Linear branched molecule consisting of decane bearing a methyl substituent at positions 4. Is is a natural product found in *Persicaria mitis*, *Persicaria hydropiperoides*, and *Persicaria* minor linear branched molecule with a non-define scent found in olive oil [46,47].
Dodecane	0.08 ± 0.36	0.11 ± 0.41	nd	0.04 ± 0.20	nd	nd	nd	nd	nd	nd	Linear branched molecule consisting of decane with 12 carbon atoms. It is a clear colorless liquid isolated from the essential oils of various plants including *Zingiber officinale* (ginger). It has a role as a plant metabolite is a natural product found in *Erucaria microcarpa*, with a balsamic scent found in olive oil [47].
Dodecane, 2,6,10-trimethyl-	nd	nd	0.04 ± 0.24	nd	0.06 ± 0.28	0.08 ± 0.10	nd	nd	nd	nd	Linear branched molecule consisting of decane with 12 carbon atoms bearing three methyl substituents at positions 2, 6 and 10. It has a role as a plant metabolite. It is a sesquiterpene with a non-define scent found in olive oil [47].
Dodecane, 4,6-dimethyl-	nd	0.05	nd	nd	nd	nd	nd	nd	nd	nd	Linear branched molecule consisting of dodecane bearing two methyl substituents at positions 4 and 6 with a non-define scent found in olive oil [47].
Heptane, 2,2,4,6,6-pentamethyl-	nd	nd	nd	nd	nd	nd	1.03 ± 0.46	nd	nd	nd	Linear branched molecule consisting of a heptane carrying two methyl groups each at positions 2 and 6, and one methyl group at position 4. is a natural product found in Tuber borchii emitted from *Tilia amurensis,* with a non-define scent found in olive oil [47].
Heptane, 3-ethyl-5-methyl-	nd	0.50 ± 0.17	nd	nd	nd	0.28 ± 1.12	nd	nd	1.33 ± 0.48	nd	Linear branched molecule of heptane carrying one ethyl and one methyl groups each at positions 3 and 5 respectively. Is a natural product found in *Tuber borchii* emitted from *Tilia amurens* with a non-define scent found in olive oil [47,48].
Hexadecane	nd	nd	0.08 ± 0.24	nd	nd	nd	nd	nd	nd	nd	Hexadecane is a straight-chain alkane with 16 carbon atoms. It is a component of essential oil isolated from long pepper. It has a role as a plant metabolite, a volatile oil component and a non-polar solvent, with a non-define scent found in olive oil [47].
Hexadecane, 2,6,10,14-tetramethyl-	nd	nd	nd	0.05 ± 0.41	nd	nd	nd	nd	nd	nd	Linear branched alkane consisting of alkane with 16 carbon atoms bearing four methyl substituents at positions 2, 6, 10 and 14 with a non-define scent found in olive oil [47].
Hexane <n->	nd	10.65 ± 0.27	nd	4.35 ± 0.06	13.40 ± 0.04	17.81 ± 0.12	nd	nd	nd	nd	Hexane is a straight-chain alkane with 6 carbon atoms. The major use for solvents containing n-Hexane is to extract vegetable oils from crops such as soybeans and in some cases olive oil with a non-define scent found in olive oil [47,49].
Nonane, 2,2,4,4,6,8,8-heptamethyl-	nd	nd	nd	nd	nd	nd	nd	nd	nd	1.46 ± 0.15	Branched alkane that is nonane carrying seven methyl substituents at positions 2, 2, 4, 4, 6, 8 and 8, odorless [50].
Nonane, 2,3-dimethyl-	nd	nd	nd	nd	nd	nd	nd	nd	nd	1.92 ± 0.06	Branched alkane consisting of nonane bearing two methyl substituents at positions 2 and 3, with a non-define scent found in olive oil [47].
Nonane, 2-methyl-	nd	nd	nd	nd	nd	0.04 ± 0.26	nd	nd	nd	nd	Methyl ketone nonane in which the methylene hydrogens at position 2 are replaced by an oxo group. Colorless to pale yellow liquid with a fruity, floral, fatty, herbaceous It has a role as a plant metabolite. Is a natural product found in *Curcuma amada, Hedychium spicatum* herbaceous odor present in olive oil found in olive oil [47].
Nonane, 2-methyl-5-propyl-	nd	nd	nd	nd	nd	nd	0.06 ± 0.48	nd	nd	nd	Linear branched alkane with 2 substituents, one methyl in position 2 and a propyl group in position 5, known as Celery ketone, with a fresh celery green cumin Odor and cumin [51].
Octane, 2,4,6-trimethyl-	nd	nd	1.06 ± 0.49	nd	nd	nd	nd	nd	nd	nd	Branched alkane consisting of decane bearing a methyl substituent at positions 2, 4 and 6, with a non-define scent [52].
Octane, 3,4,5,6-tetramethyl-	nd	nd	nd	nd	nd	nd	nd	0.56 ± 0.30	nd	nd	Branched alkane consisting of decane bearing a methyl substituent at positions 2, 4, 5 and 6 flavor and fragrance agents. Has a caramellic type odor [52].
Propane, 1-(1,1-dimethylethoxy)-2-methyl-	nd	nd	nd	2.17 ± 0.01	nd	nd	nd	nd	nd	nd	Branched alkane consisting non define scent [52].
Undecane	nd	nd	nd	nd	nd	nd	0.91 ± 0.24	nd	0.75 ± 0.07	nd	Straight-chain alkane with 11 carbon atoms appears as a colorless liquid, insoluble in water and less dense than water natural product found in *Hypericum rumeliacum*, *Persicaria mitis* and in olive oil with a faint odor [47].
Tridecane	nd	nd	nd	0.03 ± 0.13	nd	nd	nd	0.14 ± 0.35	0.11 ± 0.15	nd	Straight chain alkane containing 13 carbon atoms. It forms a component of the essential oils isolated from plants such as *Abelmoschus esculentus* and lime oil. Tridecane is a It has a role as a plant metabolite and a volatile oil component. Associated with the odor of mite-infested bin-stored wheat [47,52].
**Carboxylic acid**											
Acetic acid, 1,7,7-trimethyl-bicyclo[2.2.1]hept-2-yl ester	nd	nd	nd	0.03	nd	nd	nd	nd	nd	nd	Known as Bornyl acetate is a natural product found in *Xylopia aromatica, Eupatorium capillifolium*, and olive oil. Isolated from carrot, rosemary and sage. Flavouring agent.Pleasant, piney, balsamic odor camphor-like odor reminiscent of some varieties of pine needles and hemlock [47,53]
Acetic acid, hexyl ester	0.72 ± 0.19	0.88 ± 0.07	0.75 ± 0.57	1.22 ± 0.12	0.68 ± 0.43	0.58 ± 0.65	nd	0.11 ± 0.42	nd	0.39 ± 0.6	Hexyl acetate is the acetate ester of hexan-1-ol. It has a role as a metabolite. It is functionally related to a hexan-1-ol, is a natural product found in *Vitis rotundifolia, Lonicera japonica* and olive oil. Sweet-fruity, pearl-like odor [54]
**Alkene**											
1-Heptene, 4-methyl-	nd	nd	1.69 ± 0.55	nd	nd	nd	nd	nd	nd	nd	Linear branched alkene consisting of heptene bearing a methyl substituent at positions 4, with a non-define scent found in olive oil [47].
1-Octene, 3,7-dimethyl-	0.28 ± 0.33	nd	nd	0.48 ± 0.30	nd	nd	0.75 ± 0.01	nd	nd	nd	Linear branched alkene consisting of Octene bearing two methyl substituents at positions 3 and 7, Fragrance component woody, piney, herbaceous [55].
1-Undecene, 4-methyl-	nd	nd	nd	nd	nd	nd	nd	nd	nd	0.14 ± 0.61	Linear branched alkene consisting of undecene bearing a methyl substituent at position 4, Fragrance component with a herbaceous scent known as herbal undecanol [55].
2-Octene, 2,6-dimethyl-	nd	nd	nd	6.13 ± 0.00	nd	nd	nd	nd	nd	nd	Linear branched alkene consisting of 2-Octene bearing two methyl substituents at positions 2 and 3. Dihydromyrcenol is a monoterpenoid is a natural product found in *Vitex negundo var. cannabifolia* and *Pelargonium quercifolium* Fragance component, fresh citrus floral bergamot lime Powerful fresh Lime-like overall citrusy floral and sweet [55].
2-Undecene, 4,5-dimethyl-, [R*,S*-(Z)]-	nd	nd	nd	nd	1.62 ± 0.38	nd	nd	nd	nd	nd	Branched alkene consisting of 2-Undecene bearing a methyl substituent at positions 4 and 5 Alkene with a non-define scent [56].
5-Undecene, 9-methyl-, (Z)-	nd	nd	0.45 ± 0.60	nd	nd	nd	nd	nd	nd	nd	Branched alkene consisting of 5-Undecene bearing a methyl substituent at positions 9, responsible for the aroma components from mending yellow tea [57]
Cyclohexene, 1-methyl-4-(1-methylethylidene)-	nd	nd	nd	0.35 ± 0.10	nd	nd	nd	nd	nd	nd	Branched alkene consisting of cyclohexene carrying methyl and isopropyl substituents at positions 1 and 4 respectively. It is a monoterpene and a cycloalkene, with balsamic menthol notes [58].
**Alcohol**											
1,6,10-Dodecatrien-3-ol, 3,7,11-trimethyl-, [S-(Z)]-	nd	nd	nd	0.05 ± 0.14	nd	nd	nd	nd	nd	nd	It’s a naturally occurring sesquiterpene alcohol found in the essential oils of many types of plants. Has a waxy type odor [59].
11-Tetradecen-1-ol, (E)-	0.09 ± 0.69	nd	nd	nd	nd	nd	nd	nd	nd	nd	Alcohol with a non-define scent [46].
1-Butanol, 3-methyl-	nd	6.93 ± 0.02	12.62 ± 0.02	nd	nd	nd	nd	nd	26.07 ± 0.01	7.53 ± 0.01	Linear alcohol of 4 carbons with a methyl group in position 3 known as isoamyl alcohol common constituent of plant oils, free and as esters. Present in many wines and spirits. Flavoring agent. Present in many fruit aromas, esp. banana. Used in banana flavoring [52].
1-Heptanol, 2-propyl-	nd	nd	nd	nd	nd	nd	nd	nd	nd	0.06	It’s an alcohol with a seven-carbon chain. Musty, pungent, leafy green, with vegetative and fruity nuances of apple and banana [47].
1-Hexanol	9.56 ± 0.05	11.41 ± 0.67	nd	6.20 ± 0.06	11.41 ± 0.26	12.41 ± 0.13	11.31 ± 0.54	13.55 ± 0.11	5.63 ± 0.68	9.80 ± 0.51	It’s an organic alcohol with a six-carbon chain. Smells pungent, etherial, fuel oil, fruity and alcoholic, sweet with a green top note [60].
1-Pentanol	nd	1.59 ± 0.15	nd	nd	nd	nd	nd	nd	nd	nd	It’s an alcohol with five carbon atoms. Pungent, fermented, bready, yeasty, fusel, winey and solvent-like smell [47].
1-Propanol, 2-methyl-	nd	nd	nd	nd	nd	nd	21.51 ± 0.50	nd	nd	nd	Also called isobutanol. It’s produced by the carbonylation of propylene. Has ethereal, winey and cortex notes [60].
2-Penten-1-ol, (Z)-	1.27 ± 0.64	nd	nd	nd	nd	nd	nd	nd	nd	nd	It is a primary allylic alcohol and an alkenyl alcohol. Green notes [61].
3-Hexen-1-ol, acetate, (E)-	1.06 ± 0.14	0.75 ± 0.12	nd	1.09 ± 0.08	0.78 ± 0.06	0.61 ± 0.53	nd	0.35 ± 0.46	0.38 ± 0.46	0.44 ± 0.68	It’s a carboxylic ester. Has sharp fruity-green, green banana, pear notes [47].
3-Octanol, 3,7-dimethyl-	nd	0.19 ± 0.28	nd	nd	nd	nd	nd	nd	nd	nd	Floral linalool-like with a fatty citrus rind and tea like nuance [62].
6-Octen-1-yn-3-ol, 3,7-dimethyl-	0.08 ± 0.88	nd	nd	nd	nd	nd	nd	nd	nd	nd	Also called dehydrolinalool, is a product of linalool reduction with tropical odor [47].
Hexanol <n->	nd	nd	5.94 ± 0.61	nd	nd	nd	nd	nd	nd	nd	It’s an organic alcohol with a six-carbon chain. Pungent, etherial, fuel oil, fruity and alcoholic, sweet with a green top note [63].
**Aldehyde**											
2-Isopropenyl-5-methylhex-4-enal	nd	nd	nd	0.04 ± 0.25	nd	nd	nd	nd	nd	nd	Acyclic monoterpenoids. Powerful, herbaceous-resinous, slightly minty odor with woody-lavender-like note [64].
Hexanal	27.51 ± 0.03	31.11 ± 0.45	34.15 ± 0.05	15.56 ± 0.06	27.42 ± 0.07	29.07 ± 0.12	22.29 ± 0.17	37.57 ± 0.05	30.01 ± 0.08	28.77 ± 0.02	Also called hexanaldehyde or caproaldehyde, it is an alkyl aldehyde. Its scent resembles freshly cut grass, with a powerful, penetrating characteristic fruity odor and taste. It occurs naturally and contributes to the flavor in green peas [65].
2-Hexenal	47.54 ± 0.04	18.95 ± 0.99	28.52 ± 0.11	16.35 ± 0.04	27.31 ± 0.14	26.89 ± 0.05	13.54 ± 0.30	33.91 ± 0.04	18.78 ± 0.01	30.17 ± 0.16	2-Hexenal is a chemical compound of the aldehyde group. Imparts fresh, green, and natural top note in fruity floral types. Apple, berry, and other fruit flavors. Also, citrus flavors, especially orange juice [47].
Nonanal	0.28 ± 0.46	0.25 ± 0.22	0.26 ± 0.35	0.14 ± 0.12	0.09 ± 0.40	nd	0.14	0.17 ± 0.19	0.07 ± 0.52	0.09 ± 0.12	It’s a formally saturated fatty aldehyde resulting from the reduction of the carboxyl group of nonanoic acid. Waxy, rose and orange peel [47].
**Ketone**											
2-Oxetanone, 4-methyl-	nd	nd	nd	3.60 ± 0.09	nd	nd	nd	nd	nd	nd	Also called beta-Butyrolactone. It’s a carboxylic acid ester. Has a non-define scent.
3-Heptanone, 5-ethyl-4-methyl-	nd	9.00 ± 0.08	nd	nd	nd	nd	nd	nd	nd	nd	It’s a ketone with herbal, sweet and oily notes [66].
3-Pentanone	5.49 ± 0.11	nd	8.12 ± 0.11	nd	5.85 ± 0.05	8.91 ± 0.09	8.65 ± 0.12	10.40 ± 0.14	12.94 ± 0.01	8.80 ± 0.09	Also known as diethyl ketone, is a simple symmetrical dialkyl ketone, with an odor like that of acetone [67].
**Diene**											
1,5-Heptadiene, 2,3,6-trimethyl-	nd	nd	nd	1.95 ± 0.10	nd	nd	nd	nd	nd	nd	Branched diene carrying three methyl substituents at position 2,3, and 6. It has a non-define scent.
1,6-Heptadiene, 3,5-dimethyl-	3.12 ± 0.17	1.41 ± 0.28	nd	nd	nd	1.13 ± 0.32	nd	nd	nd	nd	Branched diene carrying two methyl substituents at position 2 and 5. It has a non-define scent.
1,6-Octadiene, 2,5-dimethyl-, (E)-	nd	nd	nd	1.43 ± 0.09	nd	nd	nd	nd	nd	nd	Branched unsaturated hydrocarbons with a non-define scent.
1,7-Nonadiene, 4,8-dimethyl-	nd	1.58 ± 0.01	nd	nd	nd	nd	1.37 ± 0.34	1.29 ± 0.36	nd	nd	Branched diene carrying two methyl substituents at position 4 and 8. It has a non-define scent.
3-Ethyl-1,5-octadiene	2.80 ± 0.34	1.33 ± 0.43	6.05 ± 0.41	0.73 ± 0.11	2.92 ± 0.01	nd	1.64 ± 0.39	1.27 ± 0.44	3.59 ± 0.05	4.59 ± 0.43	3-ethyl-1,5-octadiene is an alkadiene that is 1,5-octadiene substituted by an ethyl group at position 3. Has a non-define scent [46].
Nona-1,3,7-triene <4,8-dimethyl-, (E)->	0.12 ± 0.80	nd	nd	nd	nd	nd	nd	nd	nd	nd	Acyclic homoterpenes. Constituent of flower fragrances [68].
**Diol**											
1,7-Heptanediol	nd	nd	nd	nd	nd	nd	nd	nd	nd	5.85 ± 0.27	Diol with a non-define scent
**Ester**											
Hexyl acetate	nd	nd	nd	nd	nd	nd	nd	nd	0.35 ± 0.40	nd	Hexyl acetate is the acetate ester of hexan-1-ol. Green fruity note reminiscent of apple, pear [69].
Methyl salicylate	nd	nd	nd	0.02 ± 0.02	nd	nd	nd	nd	nd	nd	Methyl salicylate is a benzoate ester that is the methyl ester of salicylic acid. It is a colorless, viscous liquid with a sweet, fruity odor reminiscent of root beer, but often associatively called “minty” [70]
**Terpene**											
α-Muurolene	nd	nd	0.03 ± 0.01	nd	nd	nd	nd	nd	nd	nd	Belongs to the class of organic compounds known as sesquiterpenoids. These are terpenes with three consecutive isoprene units. Has woody notes [71].
α-Phellandrene	nd	nd	nd	0.29 ± 0.08	nd	nd	nd	nd	nd	nd	β-phellandrene is cyclic monoterpenes and double-bond isomers. Pleasant, fresh-citrusy and peppery-woody odor with a discretely minty note [72].
α-Pinene	nd	nd	nd	28.98 ± 0.05	3.88 ± 0.03	nd	nd	nd	nd	nd	It’s one of the lowest boiling of all monoterpenes. Intense woody, piney and terpy with camphoraceous and turpentine note. It has herbal, spicy and slightly tropical nuances [73].
β-Myrcene	nd	nd	nd	3.05 ± 0.08	nd	nd	nd	nd	nd	nd	It’s a pleasant-smelling, olefinic, acyclic unsubstituted monoterpene which occurs naturally in a large number of plant species. Herbaceous, resinous, green, balsamic, fresh hop like odor [74].
β-Ocimene	nd	nd	nd	0.08 ± 0.11	nd	nd	nd	nd	nd	nd	β-Ocimene is trans-3,7-dimethyl-1,3,6-octatriene. Exists in two stereoisomeric forms, cis and trans, with respect to the central double bond. The ocimenes are often found naturally as mixtures of the various forms. Complex note, mainly herbal lavender with green citrus, metallic and mango nuances [75].
Ɣ-Terpinene	nd	nd	nd	0.13 ± 0.17	nd	nd	nd	nd	nd	nd	It is a monoterpene and a cyclohexadiene. In gamma-terpinene the double bonds are at the 1- and 4-positions of the p-menthane skeleton. It has a characteristic lilac odor, with a sweet taste reminiscent of peach on dilution [46].
Carene <delta-3->	nd	nd	nd	nd	0.61 ± 0.13	nd	nd	nd	nd	nd	It’s a bicyclic monoterpene and is one of the components of turpentine. It has a sweet and pungent smell. It is not soluble in water, but miscible with oils and fats. Sweet, diffusive, penetrating odor, somewhat reminiscent of a refined Limonene [46].
Citronellol	nd	nd	nd	0.01 ± 0.30	nd	nd	nd	nd	nd	nd	Citronellol is a monoterpenoid that is oct-6-ene substituted by a hydroxy group at position 1 and methyl groups at positions 3 and 7. Clean, rose-like. Has a rich rosy geranium, citronella character [46].
Copaene	nd	nd	0.21 ± 0.11	0.03 ± 0.18	nd	nd	nd	nd	nd	nd	It’s an oily liquid hydrocarbon found in a number of plants that produce essential oils. Scents reminiscent of honey, spicy or woody notes [59].
3-Carene	nd	nd	nd	2.88 ± 0.08	nd	nd	nd	nd	nd	nd	3-Carene is a bicyclic monoterpene consisting of fused cyclohexene and cyclopropane rings. Carene has a sweet and pungent odor, best described as a combination of fir needles, musky earth, and damp woodlands [76].
D-Limonene	nd	0.45 ± 0.17	nd	2.54 ± 0.10	0.53 ± 0.22	nd	nd	nd	nd	nd	D-Limonene is a volatile hydrocarbon, a cycloolefin classified as a cyclic monoterpene, lemon-like odor that can be found in the rind of citrus fruits [77].

* nd = ‘not detected’, the compound was not detected in any of the three biological replicas or was found only in one of them.

## Data Availability

Not applicable.

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
