# Peer review of "Volatile Olfactory Profiles of Umbrian Extra Virgin Olive Oils and Their Discrimination through MOX Chemical Sensors"

_sensors, 2022, doi:10.3390/s22197164_

Round 1

Reviewer 1 Report

1. Some sentences have grammatically errors. Please check all text based on the grammatically structure.

2. Please check if there is a space before "in order to" in line 43.

3. the Introduction is too long. please justify novelty in Introduction.

4. What new information is obtained from the principal component analysis (PCA), and the 'Pearson correlation analyses'? The method should be introduced first in the Introduction section, the procedure described in the Materials and Methods section, and discussed in detail in the Results and Discussion section. The applicability of these methods is not clear in the manuscript. Please revise.

5. In volatile compound analysis by GC-MS, please add the calculated retention indices (RI) and retention indices taken from used reference.

6. It is not clear how the quantitative and qualitative analysis of volatile compounds is carried out, please give details.

7. Standard deviations of volatile compounds were not given and statistical analysis was not applied. Are some of the volatile compounds detected in some applications and not detected in others? The reasons for this should be explained in the text.

8. It is recommended that the observed values in the principal component analysis be changed to three experimental values.

9. Please make sure your conclusions' section underscores the scientific value-added of your paper, and/or the applicability of your findings/results. Highlight the novelty of your study.

Author Response

Thank you for your valuable comments, our response to your queries are attached in the following file

Reviewer 2 Report

This manuscript by Roberto Mariotti and colleagues present a study where several olive oil samples from an Italian region (Umbria) are studying concerning their volatile profile. Authors used several approaches, combining instrumental and sensory analyses. The topic covered by the manuscript is interesting and try to contribute for a best characterization of aroma composition of olive oils from this Italian region and respective cultivars used.

However, in my opinion, there are a few points that should be revise.

Title: Should be revise. Remove the mention “high-quality”. This is a possible conclusion (but not in this paper). Authors should emphasize the region of origin/provenience of the olive oils.

Line 15: Abstract: “EVOO” what means ? At the beginning of the abstract, authors should introduce the full name and in parentheses the abbreviation. Of course, after the abbreviation name in all manuscript.

Line 18: “gold product” ? what’s means ? this is not a scientific and technical designation. This is not supported by the results. Rewrite it.

Line 21 and table 1: Only ten samples. This is a reduce number of samples to have a correct characterization of olive oils from Umbria. Any reason for this reduces number of olive oil samples ?

Line 30: “… which faster and chipper …..” The data produced by the methodologies used didn’t introduce data that allow us this conclusion.

Line 49: “… the oil. ….” Change to “olive oil”.

Line 53: “EVOO, as a pure ‘olive juice’”. This is a popular designation (including a marketing message). Of course, this is an important source of fats with potential positive effects on human health compared to other oils, but to say that it is "pure" ....

Line 90-91: Extraction process, this is true. However, during storage several aroma profile could also be change. Rewrite and add references that support.

Line 110-117: To long sentence. Rewrite it.

Line 114: It is not clear and well explained the relevance of the use of panelist from Slow Food Organization. Why used this panelist ? and not a regional olive oil regulatory and certification federation/organization.

Table 1: The use of olive oils produces from different cultivars or blends can it not give rise to sensory profiles that can only reflect these oils and not the olive oils produced in that region with their own cultivars and influenced by the soil and climate characteristics of the region and year of production? Clarify.

Line 139-140: Not relevant for the specific objective of the paper and their results. Delete it.

Line 141- Authors are repeating several information already show in table 1. In addition, some of them should be introduce in an additional column.

Line 205-216: Taste panel: number of tasters, experience, age, sex, etc

Line 263-271: This is redundant. Include only the relevant information about statistical analysis.

Table 3: Standard deviation are missing; add a column with the sensory descriptors of each volatile compounds already described in the literature (and add the reference). This is important to increase the contribution of each volatile compound in olive oil sensory profile.

Author Response

(The authors gave the same response as above.)

Round 2

Reviewer 1 Report

Dear editor,

The revised manuscript titled “Volatile-olfactory profiles of Umbrian high-quality extra virgin olive oils and their discrimination through MOX chemical sensors” has been reviewed. In the revised version, the authors have appropriately edited and revised this earlier version according to the comments and suggestions from the reviewers, and have reasonably addressed most of the concerns and issues in the review reports. The article can be accepted for publication in Sensors.